

# A new approach for assessing climate change impacts in ecotron experiments

Inne Vanderkelen[1], Jakob Zschleischler[2,3], Lukas Gudmundsson[4], Klaus Keuler[5], Francois Rineau[6], Natalie Beenaerts[6], Jaco Vangronsveld[6,7], and Wim Thiery[1,4]

[1]Department of Hydrology and Hydraulic Engineering, Vrije Universiteit Brussel, Brussels, Belgium
[2]Climate and Environmental Physics, University of Bern, Bern, Switzerland
[3]Oeschger Center for Climate Change Research, University of Bern, Bern, Switzerland
[4]Institute for Atmospheric and Climate Science, ETH Zurich, Zurich, Switzerland
[5]Department of Environmental Meteorology, Brandenburg University of Technology Cottbus-Senftenberg, Cottbus, Germany
[6]Centre for Environmental Sciences, UHasselt, Hasselt, Belgium
[7]Department of Plant Physiology, Faculty of Biology and Biotechnology, Maria Curie-Sklodowska University, Lublin, Poland

**Correspondence:** Inne Vanderkelen (inne.vanderkelen@vub.be)

**Abstract.** Ecotron facilities allow accurate control of many environmental variables coupled with extensive monitoring of ecosystem processes. They therefore require multivariate perturbation of climate variables, close to what is observed in the field and projections for the future, preserving the co-variances between variables and the projected changes in variability. Here we present a new experimental design for studying climate change impacts on terrestrial ecosystems and apply it to the UHasselt Ecotron Experiment. The new methodology consists of generating climate forcing along a gradient representative of increasingly high global mean temperature anomalies and uses data derived from the best available regional climate model (RCM) projection. We first identified the best performing regional climate model (RCM) simulation for the ecotron site from the Coordinated Regional Downscaling Experiment in the European Domain (EURO-CORDEX) ensemble with a 0.11° (12.5 km) resolution based on two criteria: (i) highest skill of the simulations compared to observations from a nearby weather station and (ii) representativeness of the multi-model mean in future projections. Our results reveal that no single RCM simulation has the best score for all possible combinations of the four meteorological variables and evaluation metrics considered. Out of the six best performing simulations, we selected the simulation with the lowest bias for precipitation (CCLM4-8-17/EC-EARTH), as this variable is key to ecosystem functioning and model simulations deviated the most for this variable, with values ranging up to double the observed values. The time window is subsequently selected from the RCM projection for each ecotron unit based on the global mean temperature of the driving Global Climate Model (GCM). The ecotron units are forced with 3-hourly output from the RCM projections of the five-year period spanning the year in which the global mean temperature crosses the predefined values. With the new approach, Ecotron facilities become able to assess ecosystem responses on changing climatic conditions, while accounting for the co-variation between climatic variables and their projection in variability, well representing possible compound events. The gradient approach will allow to identify possible threshold and tipping points.



# 1 Introduction

Ecosystem climate change experiments are one of the key instruments to study the response of ecosystems to a change in climate. There are primarily four different factors that are altered in such experiments: temperature, precipitation, $CO_2$ concentration, and nitrogen deposition (Curtis and Wang, 1998; Rustad et al., 2001; Lin et al., 2010; Wu et al., 2011; Knapp et al., 2018). More recently multi-factor experiments have become more common. In those experiments, different combinations of the four main drivers are altered (Kardol et al., 2012; Yue et al., 2017). What is common in the majority of climate change experiments is that while the drivers of interest are being altered, all other variables are being held equal between the different treatment groups. Consequently, differences in the response can be related to the change in the main driving factor (or multiple driving factors).

In most cases, climate change experiments apply step changes to the studied drivers, that is, one factor is increased/decreased by a fixed amount. This makes it difficult to use the obtained results for model development, for which usually gradient responses need to be known. This insight has lead to the development of climate change experiments that alter the driving factors along a gradient, as implemented for instance in the Spruce and Peatland Responses Under Climatic and Environmental Change (SPRUCE, Krassovski et al., 2015) experiment. In SPRUCE spruce and peatland response to altered $CO_2$ concentration and temperature gradient are studied (no change, +4, +8, +12, and +16 degrees Fahrenheit). Implementing climate change gradients substantially reduces the number of replica per treatment and can be analyzed by a regression approach instead of an ANOVA-type of approach (Kreyling et al., 2018).

Altering only one or a limited number of climate change drivers allows for a straightforward analysis of the observed responses. However, the resulting multivariate combination of climate variables may be unphysical and may miss key aspects related to natural climate variability and driven by land-atmosphere feedbacks. For instance, droughts and heatwaves often co-occur (Zscheischler and Seneviratne, 2017) and, soil moisture conditions and precipitation occurrence are linked (Guillod et al., 2015; Moon et al., 2019). Incorporating the covariability of key climate drivers is also important for the studied responses. For instance, heatwaves characterized by similar extreme air temperatures can lead to different plant responses depending on the atmospheric conditions: under different shortwave radiation, relative humidity and surface wind conditions, the leaf temperature and the potential for heat stress varies a lot (De Boeck et al., 2016).

By focusing primarily on changes in mean climate conditions, projected change in climate variability is not taken into account in climate change experiments (Thompson et al., 2013). Changes in variability are important drivers of changes in the frequency, intensity and duration of extremes, which in turn are important drivers of ecosystem responses such as changes in community dynamics (Gutschick and BassiriRad, 2003). To capture the full range of changing climatic conditions, a holistic representation of the overall climate is necessary. In addition, other than the classic experiments with only two states (ambient versus future conditions), experiments covering a range of global warming levels allows for detection of non-linearities,



thresholds and possible tipping points, as described in the novel approach by Rineau et al. (in review).

Climate change experiments require both extensive monitoring of the ecosystem processes at various spatio-temporal scales and accurate manipulation of environmental variables to represent current and future climate conditions. Controlled envi-
ronment facilities meet these requirements by providing systems to simultaneously manipulate as well as measure multiple parameters (e.g. Lawton, 1996; Stewart et al., 2013; Clobert et al., 2018). They also allow to test the difference in response to an individual driver (e.g. one climate variable) and to simultaneous changes in multiple drivers, reflecting real-world conditions. Therefore, these types of infrastructures are very useful to perform climate change experiments, as they allow the control of a variety of climate variables with high accuracy (Rineau et al., in review). This implies that the experiments are driven by
climate forcing that represents both present and future climatic conditions in a realistic, holistic manner.

Sampling realistic climate information in a climate change context can be achieved by using climate model output. Global Climate Models (GCMs) are generally used to assess the climate state and variability at global to continental scales with a resolution of 100 to 250 km. By dynamically downscaling GCMs, Regional Climate Models (RCMs) typically resolve the
climate on a regional scale with higher spatial resolutions of 1 to 50 km. As such, RCMs allow a more realistic representation of meso-scale atmospheric processes and processes related to orography and surface heterogeneities. As climate models realistically simulate the atmospheric state under past, present and future climatic conditions with a high temporal resolution, they are suited to provide a holistic and physically consistent climate forcing for ecosystem climate change experiments. Generally, ensemble climate projections show a large spread for future climate conditions (Keuler et al., 2016), especially for variables
relevant for ecosystem experiments such as extreme temperatures, droughts and intense precipitation (Sillmann et al., 2013; Orlowsky and Seneviratne, 2013; Greve et al., 2018; Rajczak and Schär, 2017). This spread is related to (i) different climate sensitivities of the GCMs, (ii) structural differences between the models and (iii) natural variability within the climate system. The Coordinated Regional Climate Downscaling Experiment in the European domain (EURO-CORDEX) provides an ensemble of high resolution dynamically downscaled RCMs (Kotlarski et al., 2014) and is therefore highly suitable to serve as a base
for the selection of representative climate forcing for climate change experiments. With a suite of GCM/RCM combinations available, a well-informed choice on the most adequate RCM/GCM simulation can be made based on (i) the model skill in representing the observed climatology and (ii) the temperature sensitivity to future increases in greenhouse gas concentrations.

So far statistically downscaled GCM output has only rarely been used as climate forcing in ecosystem experiments. Thomp-
son et al. (2013) describe a process for generating temperature forcing for experiments in which they use daily temperature output from a GCM (MIROC) and a stochastic weather generator to generate hourly weather. They validated their method against statistical characteristics of temperature observations. Likewise, the Montpellier CNRS ecotron facility is driven by multivariate statistically downscaled GCM projections (using the ARPEGEv4 model; Roy et al. (2016). They force their experiment with climatic conditions of an average climatological year of the period 2040-2060. During the summer months,
they artificially simulate an extreme event by including a drought and heatwave by reducing the irrigation amount to zero and





increasing the air temperature artificially. However, by using a climatological year, possible extreme events are dampened by averaging. Both studies lack a thorough evaluation procedure for selecting the used climate model. Moreover, to the best of our knowledge, no study accounts for the co-variance between climate variables.

In this paper, we present a new experimental design for studying climate change impacts on terrestrial ecosystems. From an ensemble of dynamically downscaled climate model simulations, we select one simulation that well represents present-day climate conditions for four key variables in the region of interest and is representative of the multi-model mean of these variables in future projections. In this way, the new methodology accounts both for co-variance of climate parameters and for climate variability and naturally incorporates extreme events under present and future climate conditions. Furthermore, the method can

be used in a gradient approach. We apply the new methodology to generate climate forcing for the UHasselt Ecotron Experiment, an infrastructure consisting of 12 climate-controlled units, each equipped with a lysimeter containing a dry heathland soil monolith extracted from the National Park Hoge Kempen in Belgium (Rineau et al, in review). In this experiment, six units are directly forced with regional climate model output along a Global Mean Temperature (GMT) gradient anomaly.

## 2   New methodology for generating climate forcing for ecosystem climate change experiments

In our methodology, units in the ecosystem climate change experiments follow a gradient of increasing Global Mean Temperature (GMT) anomalies. In this way, a given unit is forced with the climatic conditions consistent with e.g. a 2°C warmer world, and the units represent conditions associated with increasingly warmer climates. With this approach, both the climatology and variability corresponding to these warming levels are represented. To preserve variability and co-variance between variables,

we select the best performing RCM simulation and subsequently extract the required variables from the grid cell covering the location of the experiment. By extracting a single grid cell of a single RCM simulation, climate extremes are not smoothed and the climate variability inherent to the model is fully preserved.

The methodology presented here is deployed in three steps. First, the best performing RCM projection needs to be selected

based on two criteria: (i) the simulation should have high skill in reproducing mean and extreme present-day climatic conditions and (ii) the projected future temperature anomalies should be close to the multi-model mean, that is, the selected simulation should be representative of the future mean projection (Fig. 1, step 1). To this end, the model performance is evaluated for four variables that are highly relevant for ecosystem climate change experiments: precipitation, temperature, relative humidity and surface wind speed. Precipitation is considered one of the most important variables, as water availability is likely to constrain

plant growth the most.

Second, the time windows for the different units along the GMT anomaly gradient are defined based on the annual GMT projection of the driving GCM of the chosen RCM simulation (Fig. 1, step 2). To span a large range of climate change sce-



narios, we use projections following the Representative Concentration Pathway (RCP) 8.5, a worst-case scenario following an unabated greenhouse gas emissions pathway (Riahi et al., 2011). The experiments are running for 5 years. We choose time windows corresponding to the experimental period and centred around the year in which the climatological GMT anomaly (averaged with a 30-year period) crossed the pre-defined thresholds for the first time. In the third step, the values of all neces-

5   sary variables are extracted from the chosen RCM projection based on the defined time windows for the grid cell covering the experiment location (Fig. 1, step 3). These time series are then directly used to force the ecotron units, in the highest available temporal resolution.





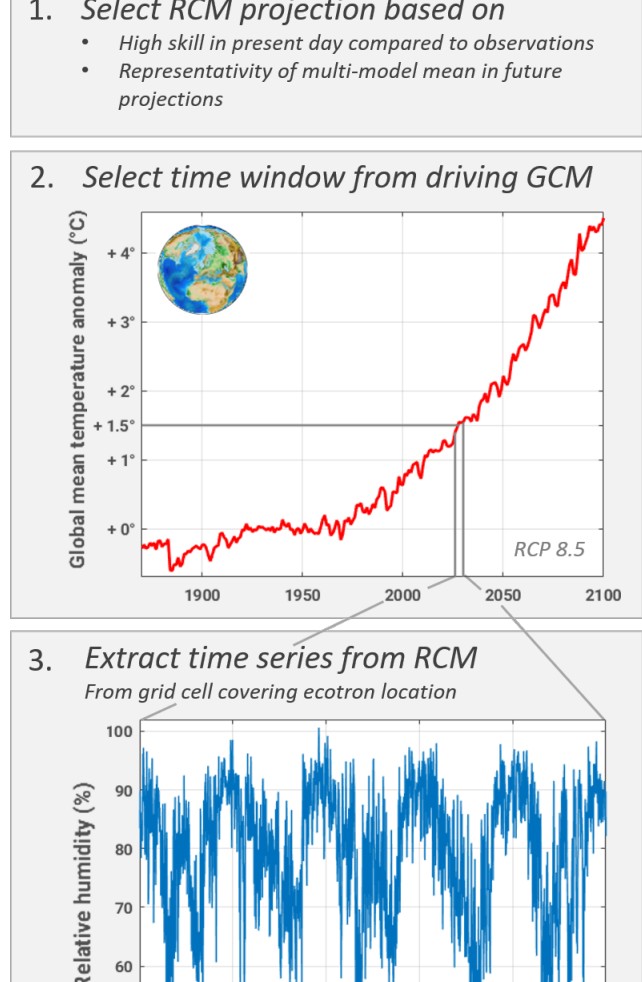

**Figure 1.** Methodology for generating climate forcing along the GMT anomaly gradient.

## 3 Data and methods

### 3.1 The UHasselt Ecotron Experiment

The UHasselt Ecotron experiment is an ecotron infrastructure consisting of replicated experimental units in which ecosystems are confined in enclosures. By allowing the simultaneous control of environmental conditions and the on-line measurement of

ecosystem processes, the ecotron units are suited for experiments with highly controlled climate change manipulation of large





intact parts of the ecosystem. The infrastructure allows an intensive monitoring and control of key abiotic parameters on 12 large-scale ecosystem replicas, called "macrocosms". These macrocosms had been extracted without disruption nor reconstitution of the soil structure from the same dry 6 to 8 years old heathland plot in the National Park Hoge Kempen (50° 59' 02.1" N, 5° 37' 40.0" E) in November 2016.

The infrastructure is a W-E oriented, 100 m by 10 m wide, and 6 m tall building (Fig. 2a). Only 12 of the 14 units are used, excluding the outermost to avoid boundary effects. Each unit consists of three compartments in which the abiotic environmental variables are controlled: the dome, the macrocosm and the chamber. The dome is transparent for photosynthetic active radiation (PAR), UVa and UVb. Here, wind and precipitation are measured and generated, and $CO_2$ , $N_2$, $CH_4$, PAR

and Net Radiation (NR; i.e. the difference in incoming and outgoing short-and longwave radiation) are measured. The second compartment, the macrocosm, contains the extracted soil column (the ecosystem) enclosed in a lysimeter. In this compartment, the soil water content, soil water tension, soil electrical conductivity and soil temperature are measured and controlled. The chamber, the third compartment, the air pressure, temperature, relative humidity, and $CO_2$ concentration are controlled (Rineau et al., in review). The ecotron infrastructure is linked with an Integrated Carbon Observation System (ICOS) ecosystem station,

which provides real-time information on local weather and soil conditions. These data are used to simulate the current weather conditions within the ecotron units (Rineau et al., in review).

The aim of the UHasselt Ecotron experiment is to study the ecological and societal impacts of climate change, by manipulating climatic variables alone or in combination and, across a wide range of predicted values, while monitoring as many soil

biota and processes as possible and to translate them into socio-economic values using heathland as a case study (Rineau et al., in review). The experiment will run uninterrupted for a period of at least five years. Six units will be used to simulate a gradient of increasing variability in precipitation regime. They are driven by the ICOS station and a perturbed precipitation time series following a gradient of increasingly long periods with no precipitation (2, 6, 11, 23, 45 and 90 days; Rineau et al, in review). In the remaining six units, atmospheric conditions along the GMT anomaly gradient will be simulated as described in section 2.

Likewise, each ecotron unit represents the local climate conditions of a globally 0° (historical), +1° (present day), +1.5° (Paris Agreement), +2°C, +3°C and +4°C warmer world. The climatology of the unit forced by +1° can thereby be directly compared to the unit driven by the ICOS station and thus representing the present-day observed conditions.



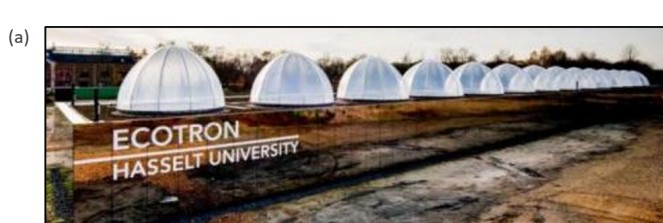

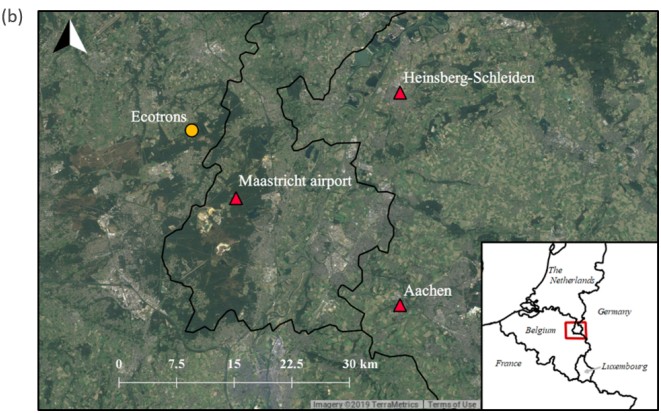

**Figure 2. The UHasselt Ecotron experiment** (a; picture: Liesbeth Driessen) and overview map with location of the infrastructure and reference weather observation stations (b).

## 3.2 Meteorological data

### 3.2.1 EURO-CORDEX

The best performing RCM simulation is selected from the Coordinated Regional Climate Downscaling Experiment in the European Domain (EURO-CORDEX), an ensemble of high resolution dynamically downscaled simulations available at a horizontal resolution of 12 km (Kotlarski et al., 2014; Jacob et al., 2014). The simulations, hereafter referred to as GCM downscalings, cover the historical period (1951-2005) and the three RCP scenarios (RCP 2.6, 4.5 and 8.5, for the period 2006-2100) by using GCMs as initial and lateral boundary conditions. Additionally, for each RCM, a reanalysis downscaling is provided in which the RCM is driven by the European Centre for Medium-Range Weather Forecasts (ECMWF) ERA-Interim as initial and lateral boundary conditions for the period 1990-2008 (hereafter referred to as reanalysis downscalings). These reanalysis-driven simulations allow to evaluate the skill of the RCMs themselves by comparing them to observations (Kotlarski et al., 2014).

In this study, we use the variables for daily mean, minimum and maximum temperature, precipitation, mean surface wind and relative humidity of all available simulations (Table 1). We consider the values of the 12 km by 12 km pixel covering the location of the reference station providing the observations. As relative humidity is not directly available for all simulations, we converted specific humidity to relative humidity using the mean temperature and surface pressure for every simulation. Com-





paring the applied conversion with the simulations for which relative humidity is available proves this conversion is applicable. Neither specific nor relative humidity are publicly available for the simulations with RegCM4-2 and ALARO-0 and the mean surface wind speed variable is not available for ALADIN53 and ALARO-0; therefore we do not analyse these variables for the respective simulations.

Once the EURO-CORDEX ensemble member is selected, the relevant variables (precipitation, mean temperature, surface pressure, surface up-welling latent heat flux and sensible heat flux, wind speed and relative humidity) are extracted from the 3 hourly RCP 8.5 simulation for the pixel covering the ecotron location for the time windows in which the GMT anomalies are crossed for each dome. These three-hourly values (except for surface up-welling latent heat flux and sensible heat flux) are then

linearly interpolated to 30 minute resolution and used to drive the climate controllers in the ecotron units. For precipitation, one additional step was added where drizzle (precipitation of less than 1 mm) was postponed and accumulated until it reached 1 mm to start a rain event in the ecotron. The surface pressure is calculated from the mean sea level pressure using the altitude of the ecotron facility (43 m a.s.l.) and assuming hydrostatic equilibrium. The concentrations of the controllable greenhouse gases ($CO_2$, $CH_4$ and $N_2O$) are determined based on the annual values calculated by van Vuuren et al. (2011) according to

RCP8.5. These correspond to the prescribed concentrations of the RCM simulations.

### 3.2.2 Weather station observations

Reference station data is obtained from the European Climate Assessment and Dataset (Klein Tank et al., 2002). The three operational weather stations closest to the UHasselt Ecotron experiment are Maastricht Airport (11km), Aachen (37km) and Heinsberg-Schleiden (29 km; Fig. 2b). These weather stations provide daily observations from the end of the 19[th] century

(Maastricht Airport and Aachen) or mid 20[th] century (Heinsberg-Schleiden) until the present-day, thereby covering both the EURO-CORDEX GCM and reanalysis downscaling periods. All stations record temperature [°C], precipitation [mm day[-1]], relative humidity [%] and surface wind speed [m s[-1]] at daily resolution, except for the Heinsberg-Schleiden station where there are no surface wind observations available.

The seasonal cycles of the observations for the different stations follow a similar annual course (Fig. 3). For temperature, the curves overlay and for precipitation they are similar. Relative humidity has a small offset between the three stations, possibly owing to the differences in absolute height and local topography. The difference in surface wind speed between Maastricht-Airport and Aachen is considerable, but is plausible considering the large spatial variability in wind speed. Given that the model evaluation showed very little sensitivity to the choice of the reference station, we hereafter present the results with the

reference station closest to the ecotron facility (Maastricht-Airport).



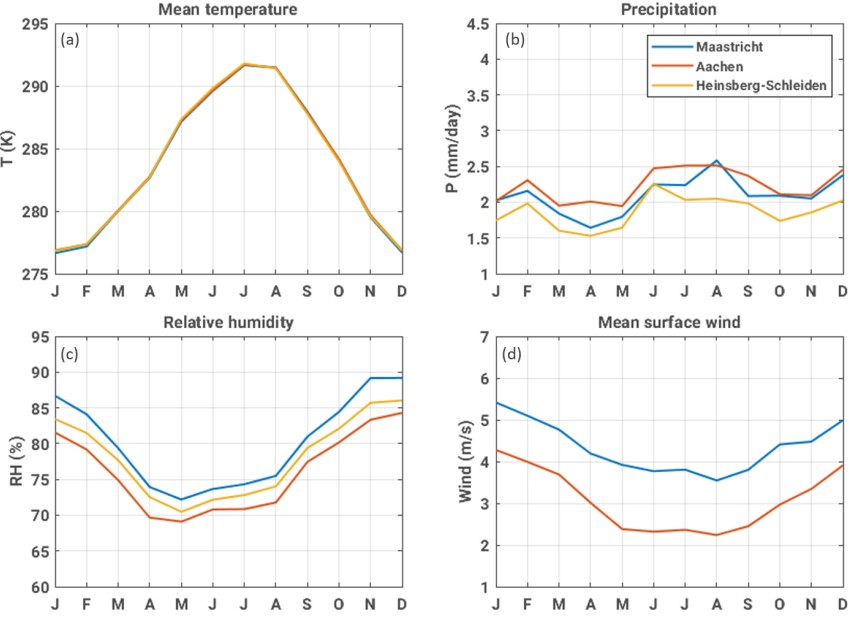

**Figure 3.** Seasonal cycles of observed mean temperature (a), precipitation (b), relative humidity (c) and mean surface wind (d) in the weather stations of Maastricht Airport, Aachen and Heinsberg-Schleiden (monthly averages based on daily data from 1963 to 2018). For Heinsberg-Schleiden no surface wind observations are available. The curves for temperature are overlaying.

## 3.3 Metrics and diagnostics

The evaluation of the EURO-CORDEX ensemble members is performed using different metrics accounting for performance of representing the climatic means, distributions and extremes.

5     A ranking is made of the reanalysis downscalings, ranging from 1-best performing model to 9-worst. First, the bias is calculated as the difference between the averages of the daily modelled and observed variables. The second metric, the Perkins Skill Score (PSS), is a quantitative measure of how well each simulation resembles the observed probability density functions by measuring the common area between two probability density functions (Perkins et al., 2007). The mean absolute error (MAE) is calculated by taking the means of the absolute differences between the modelled and observed seasonal cycles, calculated

10  based on the whole series. This is done for the whole series and to capture the potential errors in the extremes, also for the 1st, 10th, 90th and 99th percentiles which are calculated based on the daily time series of both observed and modelled time series. Next, the root mean square error (RMSE) is calculated by taking the root of the squared errors. The Spearman rank correlation (hereafter referred to as Spearman) coefficient shows the correlation of the observed and modelled series, calculated based on daily values. Finally, the Brier Skill Score (BSS) is calculated, which gives an indication of the improvement of the Brier Score

15  (an index to validate probability forecasts) compared to a background climatology in which each event has an equal occurrence





probability (Brier, 1950; Murphy, 1973). For the GCM downscalings, we use the same ranking method and scores, except for the RMSE, Spearman rank correlation and BSS because the internal variability, inherent to individual simulations with a coupled climate model, can not be predicted on multi-decal timescales, and can therefore not be compared to observations on a day-by-day basis (Fischer et al., 2014; Meehl et al., 2014).

In addition to the performance metrics computed on the actual time series, the RCM performance is also evaluated based on the bias in climatological diagnostics related to temperature and precipitation. To this extent, the average diurnal temperature range (DTR [K]; the difference between the daily maximum and minimum temperature) is calculated for the whole year, for the winter (December-January-February) and summer (June-July-August) season. Next, the number of wet days (defined as

days during the year for which precipitation is larger than 0.1 mm or larger than 1 mm) and the number of frost days (days with a minimum temperature below 0°C) are calculated. Furthermore, the monthly maximum 1-day precipitation (Rx1day [mm day$^{-1}$]) and the number of consecutive dry days (CDD [days]; the annual maximum number of days for which precipitation is below 1 mm) and consecutive wet days (CWD [days]; the annual maximum number of days for which precipitation is equal to or more than 1 mm) are included in the analysis. All indices are calculated for the simulated and observed time

series, and consequently the ranking is established based on the difference between the model and observed diagnostic. Next, the correlation between the different variables is evaluated by comparing them to the observed correlation. This is done both on annual time scale and for the summer and winter seasonal averages, as correlations are expected to differ in sign and magnitude between the two seasons (e.g. negative correlation between temperature and relative humidity in summer reflecting heatwave conditions, and a positive correlation between wind speed and precipitation in winter reflecting storm conditions).

After choosing the best performing simulation based on the evaluation of both the reanalysis and GCM downscalings, the climate change signals for this simulation are investigated by calculating changes in various climate change indices, based on the Expert Team on Climate Change Detection and Indices (ETCCDI; see http://etccdi.pacificclimate.org/list_27_indices. shtml) for the 5-year periods defined by the GMT anomalies relative to the reference period (1951-1955). These indices are

widely used for analyzing changes in extremes (e.g. Zhang et al., 2009; Orlowsky and Seneviratne, 2013; Sillmann et al., 2013). The temperature indices are (i) $\Delta T$ [°C], the mean daily temperature change, (ii) $\Delta TXx$ [°C], the difference in annual maximum value of daily maximum temperature,(iii) $\Delta TNn$ [°C], the difference in annual minimum value of daily minimum temperature, (iv) $\Delta$ frost days, the difference in number of frost days (with a minimum temperature below 0°C), (v) $\Delta$ summer days, the difference in number of summer days (with the maximum temperature above 25°C), and finally (vi) $\Delta GSL$ [days],

the difference in growing season length, defined as the annual count between the first span of at least 6 days with a daily mean temperature higher than 5°C and the first span after July 1st of 6 days with a daily mean temperature lower than 5°C. The precipitation indices are (i) $\Delta PRCPTOT$ [mm], the difference in annual accumulated precipitation (as simulated over the five-year period), (ii) $\Delta Rx1day$ [mm] the difference in monthly maximum 1-day precipitation, (iii) $\Delta R10mm$ [days] the difference in number of days per year with more than 10 mm precipitation, (iv) $\Delta CDD$ [days] the difference in the maximum

length of a dry spell (measured as the maximum number of consecutive days with less than 1 mm precipitation) and finally, (v)





$\Delta CWD$ [days] the maximum length of a wet spell (measured as the maximum number of consecutive days with more than 1 mm precipitation).

### 3.4 Applying the new methodology for the UHasselt Ecotron experiment

The best performing RCM simulation is identified by elimination based on expert judgment based on the performance of the two selection criteria. Next, we define the time windows for the different units along the gradient based on the 30-year averaged GMT anomaly of the driving GCM under RCP8.5 relative to 1951-1955 (Section 2, Fig. 1, table 2). Based on these time windows, we extract the three-hourly data for all necessary variables from the simulation for the 11 km by 11 km grid cell covering the location of the experiment.

## 4 Results

### 4.1 Identification of the best performing model simulation

#### 4.1.1 First criterion: skill in present-day climate

Overall, model skill strongly varies across RCMs (Fig. 4). While the annual temperature cycle is generally well represented by all RCMs, biases may reach up to 2 degrees in individual months for some RCMs. The biases in precipitation are generally positive (up to factor 2.4) and vary across RCMs. Only CCLM4-8-17 simulates precipitation in the same range as the observed climatology (nearly no bias (100.22%) on annual mean precipitation amounts), while the other RCMs overestimate the total precipitation amounts from 114% up to 182%. For relative humidity and surface wind speed, all RCMs generally succeed in representing the seasonal cycle, but exhibit deviations in amplitude and absolute values (e.g. amplitude biases of RCA4 (-37.8%), ALADIN53 (23.3%) and CCLM4-8-17 (+16.3%) for relative humidity, and annual mean biases for WRF331F (+15.6%) and HIRHAM5 (-9.1%) for surface wind speed). Overall, these seasonal cycles indicate that for all simulations, the relative bias in precipitation is large compared to biases in other variables.

The rankings of the reanalysis downscalings for the four variables (Fig. 5) indicate that, overall, CCLM4-8-17, RACMO22E, REMO2009 and HIRHAM5 are performing best. CCLM4-8-17 and RACMO22E show the highest relative skill for precipitation, while REMO2009 and HIRHAM5 demonstrate high skill for temperature. CCLM4-8-17 is the best performing model based on the bias and total MAE metrics for temperature and precipitation, but is ranked in the mid range for the metrics related to the shape of its temperature distribution (PSS and percentile MAE). This can be attributed to an overestimation of the amplitude of the seasonal temperature cycle in this model (too cold in winters, too hot in summers; Fig. 4a, (Kotlarski et al., 2014). For relative humidity and surface wind speed, RACMO22E generally demonstrates the highest skill. Considering the climatological diagnostics (Fig. 7a), CCLM4-8-17 shows the highest relative skill for precipitation-related diagnostics (wet days, monthly maximum 1-day precipitation, length of dry and wet spells), while RACMO22E and RCA4 show higher relative




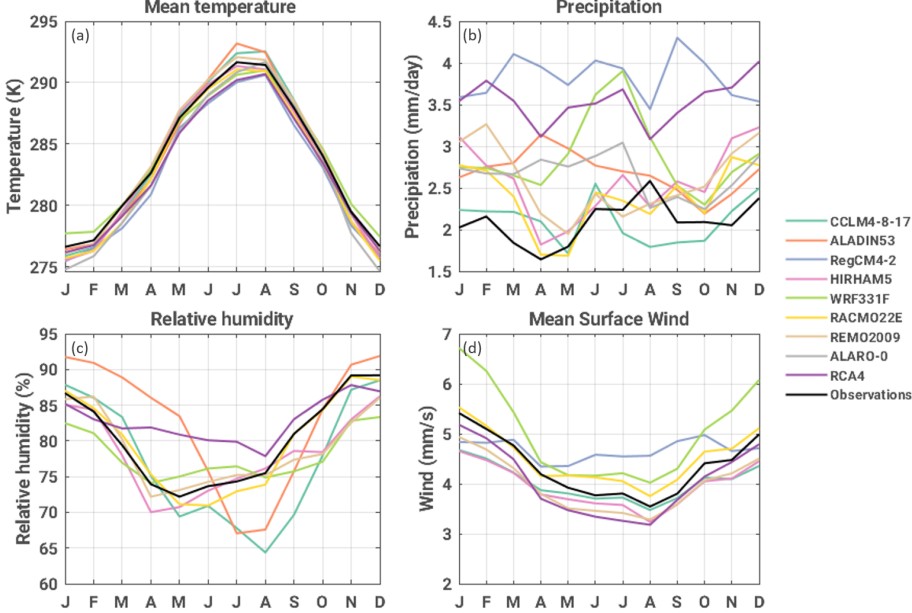

**Figure 4. Seasonal cycle of the reanalysis downscalings** for mean temperature (a), precipitation (b), relative humidity (c) and mean surface wind speed (d). (The RegCM4-2 and ALARO-0 simulations are not available for relative humidity and the ALADIN53 and ALARO-0 simulations are not available for surface wind speed.)

skill for the annual, winter and summer diurnal temperature range. While RCA4 is highly ranked for temperature-related diagnostics, it is one of the models with the lowest relative skill for precipitation-related diagnostics. The correlation ranking shows a more scattered image, for the annual correlation as well as summer and winter correlations (see appendix Fig. A2). Overall, as the reanalysis driven simulations with ALADIN53, RegCM4-2, WRF331F and ALARO-0 show the lowest skill compared to the other RCMs, we take them out of consideration to serve as ecosystem forcing.

Second, we evaluate the GCM downscalings for the period 1951-2005. The seasonal cycles of the temperature, precipitation, relative humidity and surface wind speed show a similar pattern as the reanalysis downscalings, with again a strong wet bias for precipitation in most models (see appendix Fig. A1). The rankings show a mixed pattern for the different variables: there are no simulations which rank high for all considered variables ( Fig. 6). For precipitation, the simulations with CCLM4-8-17, RACMO22E have better relative skill compared to the other simulations, which is in line with the high ranking of these models in the reanalysis downscalings. Furthermore, it is remarkable that the simulations which show a high skill for precipitation, typically show lower skill for relative humidity and vice versa, e.g. CCLM4-8-17 driven by HadGEM2-ES (high ranking in precipitation, lowest in relative humidity) and REMO2009 driven by MPI-ESM-LR (high ranking in relative humidity and lower in precipitation). The three MPI-ESM-LR driven simulations appear to be better in reproducing the temperature climatology compared to the other simulations. For the climatological diagnostics, generally CCLM4-8-17 is scoring best for the

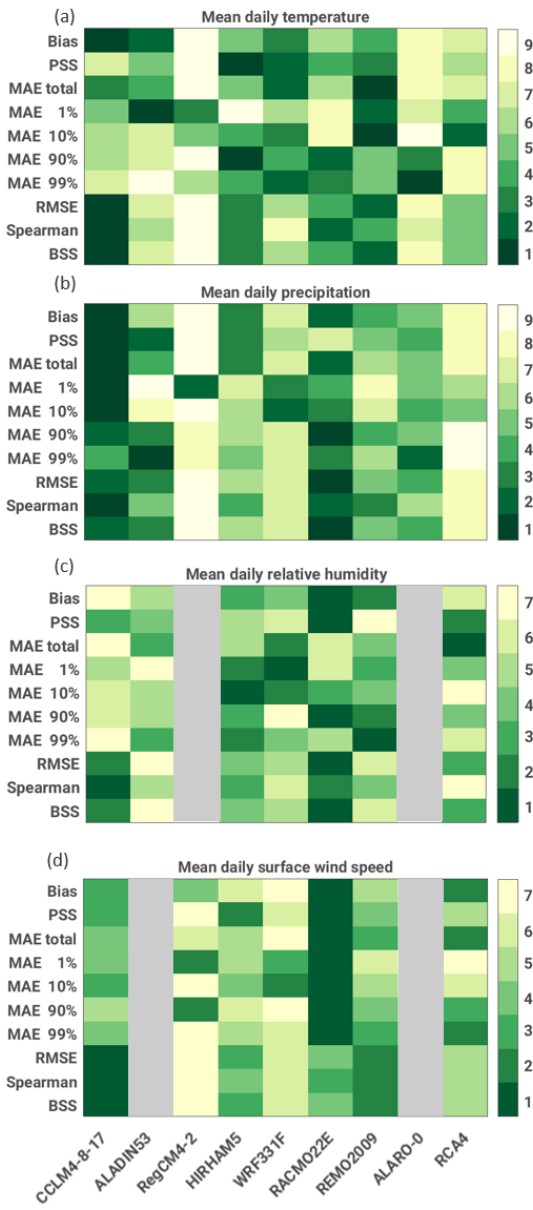

**Figure 5. Ranking of the reanalysis downscalings** based on performance on temperature (a), precipitation (b), relative humidity (c) and surface wind speed (d) compared to observations from Maastricht. The metrics shown are the Bias, Perkins Skill Score (PSS), Mean Absolute Error (MAE) for the entire time series and the 1st, 10th, 90th and 99th percentiles, Root Mean Square Error (RMSE), Spearman rank correlation (Spearman) and Brier Skill Score (BSS). Rankings are from 1-best to 9-worst. Grey colors indicate that the variable is not available for the considered model.





precipitation-related diagnostics, whereas simulations with RCA4 are ranked the best for DTR (annual, summer and winter).

Based on the ranking of the GCM downscalings, the following simulations are considered potential candidates to serve as climate forcing: CCLM4-8-17 driven by CNRM-CM5, EC-EARTH and MPI-ESM-LR, HIRHAM5 driven by EC-EARTH
and HadGEM2-ES, and RACMO22E driven by HadGEM2-ES (Figs. 5,6 and 7). Since precipitation biases strongly differ among RCMs (table 1), and since precipitation is a critical variable for the ecosystem experiments (Van der Molen et al., 2011; Vicca et al., 2014; Estiarte et al., 2016), we prioritize a minimum relative bias for precipitation over a lower bias for temperature, relative humidity and surface wind speed. The precipitation biases for the considered simulations are +150 mm year$^{-1}$ for CCLM4-8-17 driven by CNRM-CM5, +8 mm year$^{-1}$ for CCLM4-8-17 driven by EC-EARTH, +24 mm year$^{-1}$ for
CCLM4-8-17 driven by MPI-ESM-LR, +323 mm year$^{-1}$ for HIRHAM5 driven by EC-EARTH, 101 mm year-1 for HIRHAM5 driven by HadGEM2-ES and 35.51 mm year$^{-1}$ for RACMO22E driven by HadGEM2-ES. Based on this, the CCLM4-8-17 EC-EARTH driven simulations has the best chance to be chosen as forcing, followed by the CCLM4-8-17 MPI-ESM-LR and the RACMO22E HadGEM2-ES driven simulation.

**4.1.2   Second criterion: Representativeness of multi-model mean in future projections**

To verify the second requirement we look at anomalies from the mean signal of the four variables for the future period of the simulations under RCP 8.5. The EC-EARTH driven CCLM4-8-17 simulation is representative of the multi-model mean for all four variables (Fig. 8), and even the median simulation for the mean temperature anomaly. For precipitation and relative humidity however, the CCLM4-8-17 EC-EARTH simulation show decreasing anomalies after 2050. underestimates the multi-
model mean anomaly. The other selected simulations have a larger positive bias in precipitation for their GCM downscalings. A possible reason is that these simulations overestimate precipitation and simulate a more intensive hydrologic cycle, which also implies stronger changes in the future.

The remaining five simulations from step 1 (CCLM4-8-17 driven by MPI-ESM-LR, HIRHAM5 and RACMO22E driven by
HadGEM2-ES) all systematically underestimate or overestimate other variables (Figs. A4,A5, A6, A7 and A8). For instance, the mean temperature anomaly of CCLM4-8-17 driven by MPI-ESM-LR simulation (1.46 °C) is lower than the 10$^{th}$ percentile of all simulations (1.51 °C) and the temperature anomaly for CCLM4-8-17 driven by CNRM-CM5 is the 30$^{th}$ percentile (1.67 °C). HIRHAM5 driven by HadGEM2-ES overestimates relative humidity anomalies compared to the multi-model mean, with a mean value (1.26 %) around the 80th percentile. Finally, the HadGEM2-ES driven RACMO22E simulation overestimates rel-
ative humidity and temperature anomalies, up to the 90th percentile for temperature. Overall, we conclude that the EC-EARTH driven CCLM4-8-17 simulation is the most appropriate candidate for serving as climate forcing for the UHasselt Ecotron experiment.





**Table 1.** Bias in annual precipitation ($P$ bias) and rank based thereof (from 1-best to 18-worst) for the EURO-CORDEX GCM downscalings for the period 1951-2005 over Maastricht-Airport.

| RCM | GCM | $P$ bias (mm/year) | Rank |
|---|---|---|---|
| CCLM4-8-17 | CNRM-CERFACS-CNRM-CM5 | 145 | 8 |
| CCLM4-8-17 | ICHEC-EC-EARTH | 8 | 1 |
| CCLM4-8-17 | MOHC-HadGEM2-ES | -174 | 9 |
| CCLM4-8-17 | MPI-M-MPI-ESM-LR | 24 | 2 |
| ALADIN53 | CNRM-CERFACS-CNRM-CM5 | 550 | 14 |
| HIRHAM5 | ICHEC-EC-EARTH | 323 | 12 |
| HIRHAM5 | MOHC-HadGEM2-ES | 101 | 6 |
| HIRHAM5 | NCC-NorESM1-M | 571 | 16 |
| WRF331F | IPSL-IPSL-CM5A-MR | 726 | 18 |
| RACMO22E | ICHEC-EC-EARTH | 99 | 5 |
| RACMO22E | MOHC-HadGEM2-ES | 36 | 3 |
| REMO2009 | MPI-M-MPI-ESM-LR | 225 | 10 |
| ALARO-0 | CNRM-CERFACS-CNRM-CM5 | 560 | 15 |
| RCA4 | CNRM-CERFACS-CNRM-CM5 | 319 | 11 |
| RCA4 | ICHEC-EC-EARTH | 386 | 13 |
| RCA4 | IPSL-IPSL-CM5A-MR | 691 | 17 |
| RCA4 | MOHC-HadGEM2-ES | 111 | 7 |
| RCA4 | MPI-M-MPI-ESM-LR | 70 | 4 |



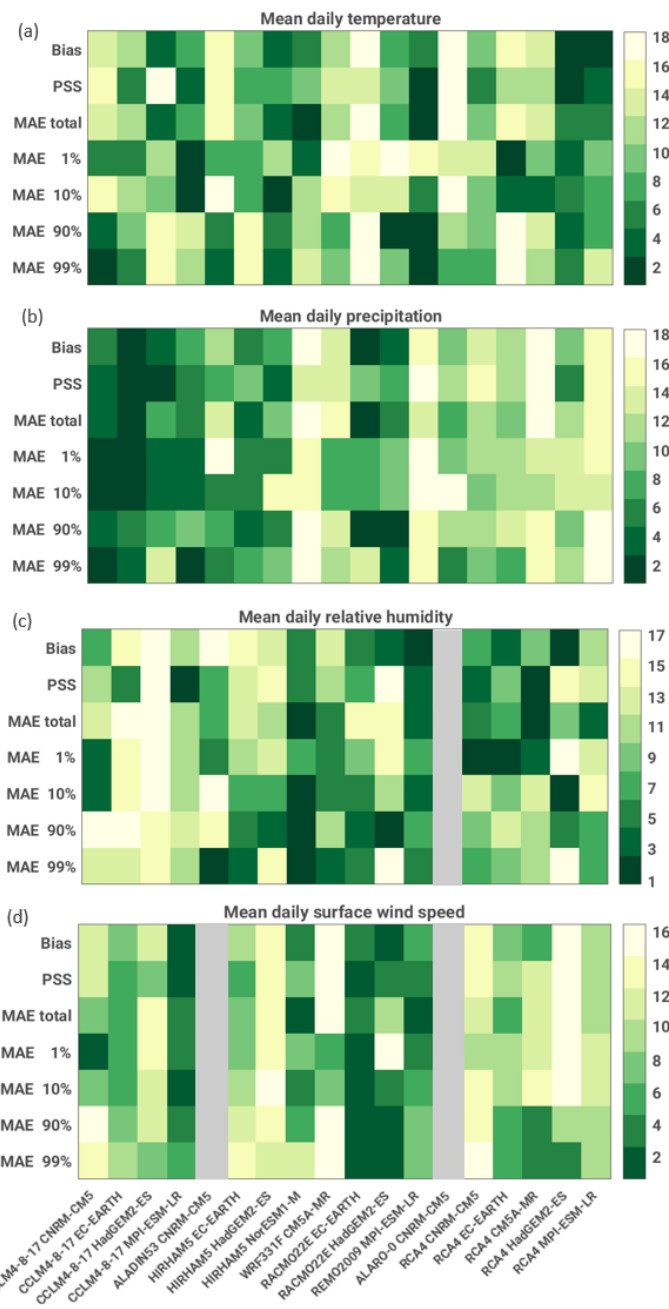

**Figure 6. Ranking of the GCM downscalings** based on performance on temperature (a), precipitation (b), relative humidity (c) and surface wind speed (d) compared to observations from Maastricht. The metrics showed are the bias, Perkins Skill Score (PSS), Mean Absolute Error (MAE) for the total and 1st, 10th, 90th and 99th percentile. Rankings are from 1-best to 16, 17 or 18-worst for surface wind speed, relative humidity, precipitation and temperature, respectively. Grey colors indicate that the variable is not available for the considered model.





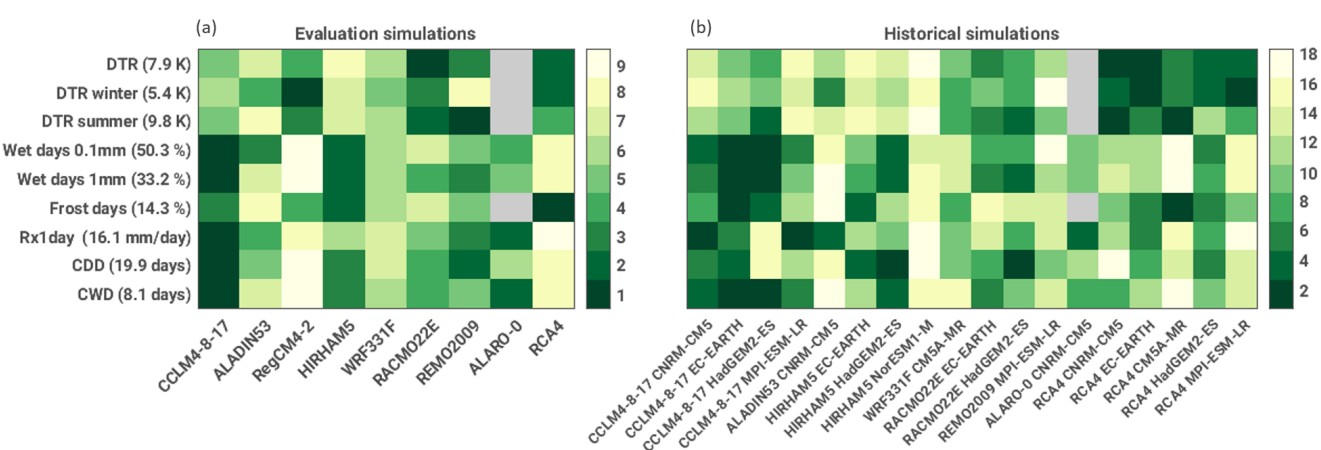

**Figure 7. Ranking of the reanalysis (a) and GCM (b) downscalings for the historical period based on climatological diagnostics.** Diurnal temperature range (DTR) in summer (July-August) and winter (December-February), number of wet days defined as days with precipitation > 0.1 mm and precipitation > 1 mm, number of frost days defined as days with mean temperature < 273.15 K, Monthly maximum 1-day precipitation (Rx1day), consecutive dry days (CDD), the maximum length of a dry spell, and consecutive wet days (CWD), the maximum length of a wet spell. Next to the diagnostic name its value as observed in Maastricht-Airport is shown. Rankings are from 1-best to 9 or 18-worst for the reanalysis and GCM downscalings, respectively.





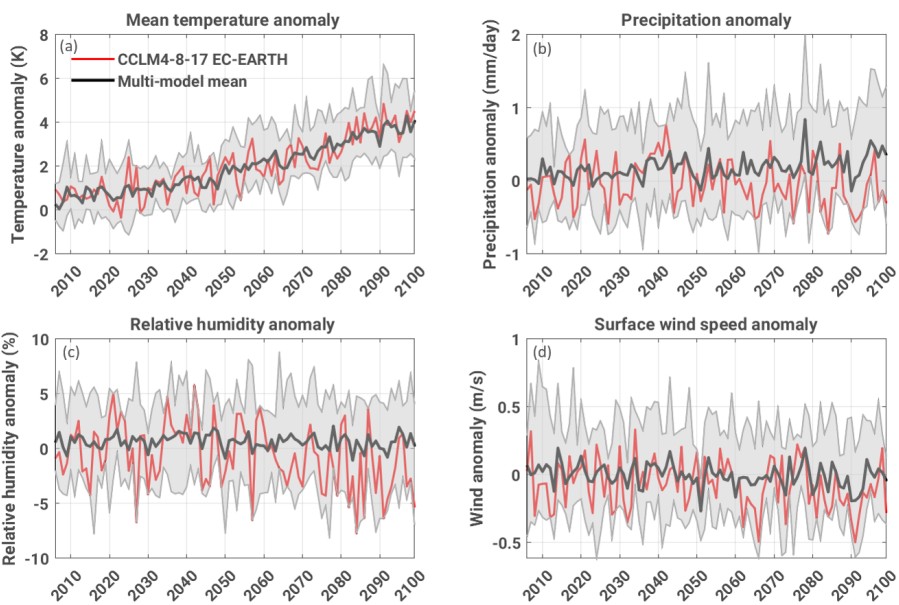

**Figure 8. Anomalies for the CCLM4-8-17 EC-EARTH simulation following RCP 8.5** at the ecotron site for temperature (a), precipitation (b), relative humidity (c) and surface wind speed (d). The reference period is 1977 to 2006, the anomalies of the CLM4-8-17 EC-EARTH simulation are calculated compared to its own values in the reference period. In gray the envelope of all EURO-CORDEX RCP8.5 simulations is showed.





## 4.2 Characterization of the selected meteorological forcing

Based on the selection criteria we single out the EC-EARTH (ensemble member r12i1p1) driven CCLM4-8-17 simulation as climate forcing for the UHasselt Ecotron experiment. The climatic conditions in the six units along the gradient represent an increasing signal of climate change. The overall trend of the local temperature anomaly compared to the reference period (0°C)

increases monotonically with the corresponding GMT anomalies (Fig. 9a). No clear trends are visible for precipitation, relative humidity and surface wind speed anomalies, but very clear for the minimum and maximum temperature anomalies which are both increasing (Fig. 9). The mean daily temperature is increasing at a similar rate compared to GMT anomaly, and minimum and maximum temperature show a larger increase (table 2). None of the temperature indices show a linear increase, reflecting the difference between global and local climatic conditions and the influence of decadal internal variability. The ecotron

unit representing a +4°C world is the most extreme case, with increases of $TXx$ of +6.30 °C and an increase of $TNn$ with +10.21°C (table 2). The number of frost days decreases with about -76.2, while the number of summer days with a temperature above 25° C increases with about 36.6 days. The annual growing season length is extended with 80 days on average, leaving only 59.4 days of the year not favourable for growth. The indices for precipitation show a less clear trend (table 2). The total precipitation amount varies for the five units, without any trend and shows a substantial decadal variability in all seasons (see

Fig.9) . $Rx1day$ has positive anomalies for the +1.5°C, +2°C and +3°C units (+0.35 mm day$^{-1}$ +1.92 mm day$^{-1}$ and +2.34 mm day$^{-1}$, respectively). These +2°C and +3°C units also knows an increase in $R10mm$ (+3.2 and +3.6 days) compared to the other units. Finally, there is no clear trend in $CWD$, but there is an increase in $CDD$ up to +11.8 days for the +4°C unit. The +1.5°C unit spans a drier time window, with an average $CDD$ of +9.6 days. Figure 9 further shows a systematic decrease of relative humidity during summer with increasing warming and a strong decadal variability of surface wind speed especially in

winter.



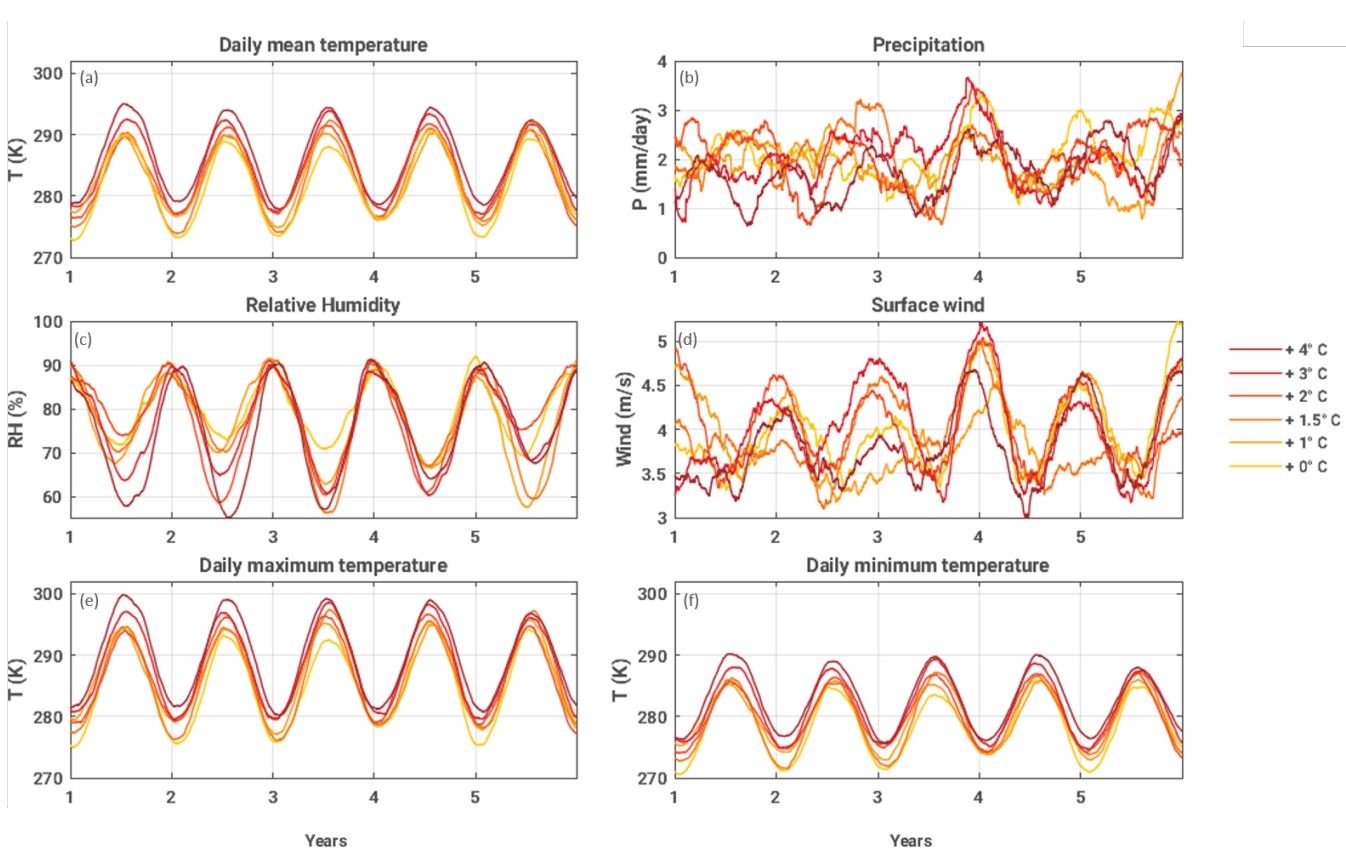

**Figure 9. Annual cycles of the CCLM4-8-17 EC-EARTH ecotron unit forcing** for the +1°C, +1.5°C, +2°C, +3°C and +4°C units compared to the 0°C reference period. Curves were smoothed using Savitzky-Golay filtering (order = 2 frame = 301; Savitzky and Golay (1964)

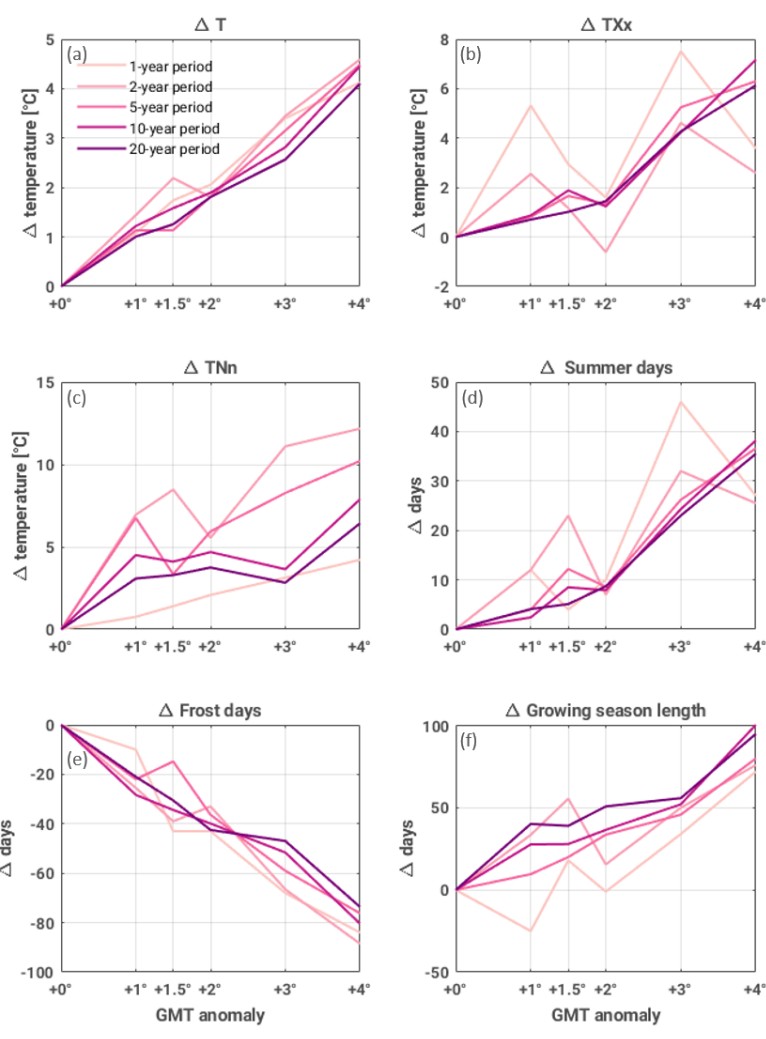

**Figure 10. Annual anomalies per GMT anomaly for increasing time window lengths (ranging from a 1-year period to a 20-year period) of the CCLM4-8-17 EC-EARTH simulation following RCP 8.5** for temperature indices: mean temperature anomaly ($\Delta T$; a), annual maximum temperature ($\Delta TXx$; b), annual minimum temperature ($\Delta TNn$; c); anomaly in annual number of summer days (d), frost days (e) and the anomaly in growing season length (f). Note the different y-axis scales.

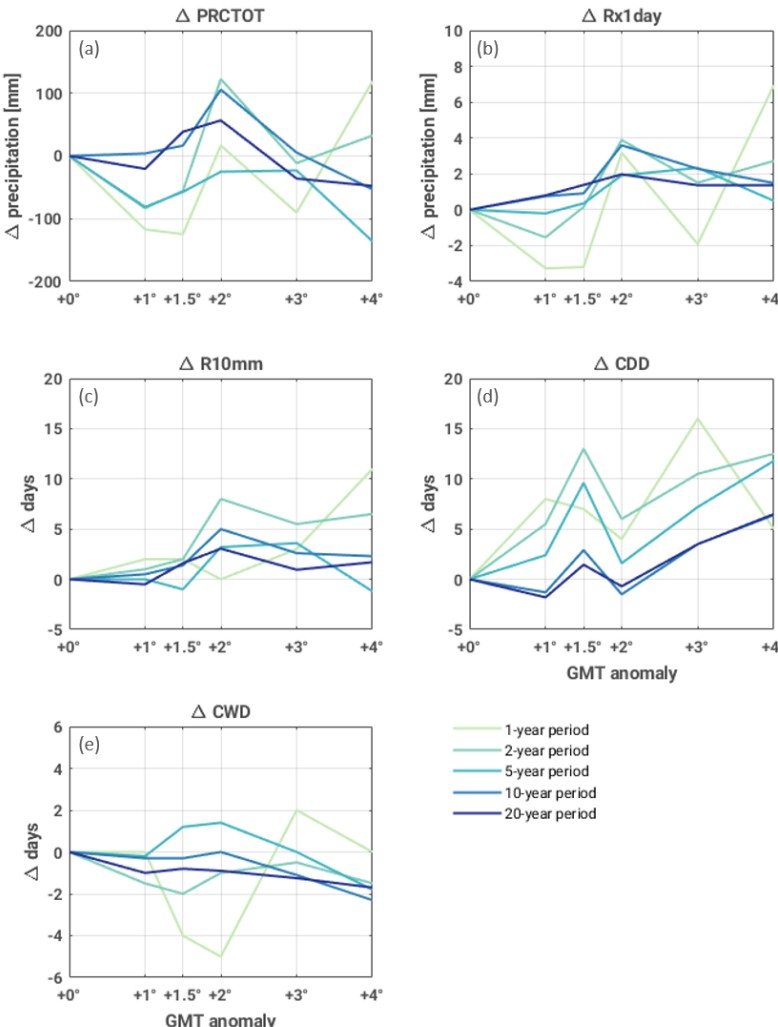

**Figure 11.** Same as Fig. 10, but now for precipitation indices: the annual accumulated precipitation anomaly ($\Delta PRCPTOT$; a), anomaly of monthly maximum 1-day precipitation ($\Delta Rx1day$; b), anomaly of annual number of days with more than 10 mm precipitation ($\Delta R10mm$; c), anomaly of annual maximum length of a dry spell ($\Delta CDD$; d) and anomaly of maximum length of a wet spell ($\Delta CWD$; e). Note the different y-axis scales.



**Table 2. Extracted 5-year periods and temperature and precipitation indices based on ETCCDI for the CCLM4-8-17 EC-EARTH simulation at the ecotron location.** The 0°C column gives the absolute reference values. The periods are calculated based on the 30-year averaged global mean temperature (GMT) anomaly calculated from EC-EARTH.

|  | 0 °C *(ref value)* 1951 - 1955 | +1 °C 2011 - 2015 | +1.5 °C 2028 - 2032 | +2 °C 2043 - 2047 | +3 °C 2067 - 2071 | +4 °C 2091 - 2095 |
|---|---|---|---|---|---|---|
| $\Delta T$ [°C] | *8.17* | +1.13 | +1.14 | +1.81 | +3.15 | +4.49 |
| $\Delta TXx$ [°C] | *30.98* | +0.82 | +1.66 | +1.34 | +5.24 | +6.30 |
| $\Delta TNn$ [°C] | *-12.73* | +6.75 | +3.34 | +5.94 | +8.27 | +10.21 |
| $\Delta$ Frost Days | *103* | -22 | -14.8 | -36.4 | -59 | -76.2 |
| $\Delta$ Summer Days | *11.4* | +4 | +12.2 | +8.6 | +26.2 | +36.6 |
| $\Delta GSL$ [days] | *225.6* | +9.6 | +20 | +33.6 | +45.8 | +80 |
| $\Delta PRCPTOT$ [mm] | *771.09* | -81.32 | -57.2 | +25.12 | -23.14 | -136.05 |
| $\Delta Rx1day$ [mm] | *14.38* | -0.2 | +0.35 | +1.92 | +2.34 | +0.5 |
| $\Delta R10mm$ [days] | *14.6* | 0 | -1 | +3.2 | +3.6 | -1.2 |
| $\Delta CDD$ [days] | *17.2* | +2.4 | +9.6 | +1.6 | +7.2 | +11.8 |
| $\Delta CWD$ [days] | *9.6* | -0.2 | +1.2 | +1.4 | 0 | -1.8 |





## 5   Discussion

The presented methodology exhibit some challenges, which are addressed in the following section.

We extract all climate variables from one grid cell of the RCM simulation to conserve a realistic, non smoothed signal.
However, the extracted time series of the grid cell can differ a lot between different models and time periods, reflecting the natural climate variability. GCMs and RCMs provide robust signals when aggregated over a larger spatial area (Seneviratne et al., 2016; Fischer and Knutti, 2015). By taking the spatial mean, a more robust estimate of the mean climate is obtained, including robust signals of climate change. This explains the difference in local climate change signals (Fig. 8, table 2) and non-linearities compared to the GMT anomaly obtained by global averaging (Seneviratne et al., 2016). It is however necessary
to use actual time series from a single grid cell to capture e.g. the extreme precipitation event occurring in the considered grid cell, but not in the neighbouring grid cells. The grid-cell values also reflect strong interannual to decadal variability which is of high relevance for a realistic forcing of the ecosystem.

Climate model simulations are often biased, which is mostly related to structural model deficiencies (Flato et al., 2013).
Applying bias adjustment is a standard way to deal with biases (Gudmundsson et al., 2012; Vanderkelen et al., 2018), but such methods face several challenges and need to be chosen carefully to not increase biases in the co-variability of variables (Zscheischler et al., 2019). In the proposed method we therefore directly use the 'raw' model output, as such preserving climate variability and the physically-consistent co-variance of the different meteorological variables. In this way, the Ecotron experiment will study ecosystem responses to multi-variate drivers as compound controls. For instance, it will provide a unique
opportunity to study the impact from realistic compound events (Zscheischler et al., 2018), e.g. events similar to the drought-heat event of 2018, which caused massive heather die-off both in the field and in the ecotrons, forced by conditions like they happened in the field.

The gradient for the different ecotron units does not follow a monotonic trend for some of the key indicators (Fig. 9 and
table 2), due to the high local and inter-annual natural climate variability of the climate system. This issue could be alleviated by running the experiment for a longer period. Comparing different time frames, all extracted based on 30-year averaged GMT anomaly thresholds, shows that choosing longer time windows of 10 or 20 years leads to more clear monotonic trends (Figs. 10 and 11), which is more pronounced for temperature-derived indices than for precipitation-derived indices. For shorter time windows of 1 to 2 years, the inter-annual and local natural variability leads to larger variations in trend for the different GMT
anomaly levels. Therefore, the experiment would have to run for a long period, but the experimental time frame is constrained by the experimental setup and possible renewal. As a compromise, here we use a 5-year experimental period. Ideally, the entire gradient should be replicated several times with different climate trajectories to average out the natural climate variability. This approach is however constrained by the high cost of the experimental set-up.



In the different ecotron units, we assume that the controlled variables ($CO_2$ and $CH_4$ concentration, temperature, precipitation, atmospheric humidity, wind, ...) are in equilibrium with the warming level, by extracting the 5-year period in which the GMT anomaly in the driving GCM is reached. While this is a reasonable assumption, several components in the climate system will not yet be in equilibrium with the GMT anomaly at the time of simulation (e.g. glaciers, ice sheets, sea level; Zekollari et al. (2019), Church et al. (2013). Therefore, we cannot rule out that changes in these slower components may still affect the meteorological conditions until these reach equilibrium too. For instance, a delayed melting of sea ice could alter the polar circulation and thereby affecting the mid-latitude circulation (Coumou et al., 2018), whereas ice sheet melting may affect oceanic pole-ward heat transport (Caesar et al., 2018). However, to select the time windows, we follow the same approach as the Transient Response to Cumulative Emissions (TRCE) as presented in the Intergovernmental Panel on Climate Change (IPCC) Fifth Assessment Report (IPCC 2013, 2013). This concept describes the warming per unit of carbon emissions, which largely follows a linear relationship independent of the emission scenario (Knutti and Rogelj, 2015).

Finally, the set-up of the UHasselt Ecotron experiment implies that the incoming shortwave radiation will follow current weather conditions and not the weather conditions as prescribed by the RCM forcing. It is thus possible to have, for instance, clear-sky conditions and associated high incoming shortwave radiation in the field, while in the ecotron unit a heavy precipitation event is simulated consistent with the RCM forcing. In this example, the system receives more incoming shortwave radiation than in the simulated climate. Likewise, the surface fluxes will be higher, but the resulting temperature and moisture is corrected within the ecotron unit by the controlling devices to fully follow the boundary layer conditions as they are prescribed by the RCM.

The UHasselt Ecotron experiment allows to investigate ecosystem responses to different levels of climate change. This allows to study subtle changes in ecosystem responses such as impacts of decreased frost frequency on plant mortality (Berendse et al., 1994) and the interactions between the occurrence of mild droughts and plant acclimation for longer droughts (Backhaus et al., 2014). Although climate variables are prescribed, ecosystem-climate feedbacks originating from interactions between the biosphere and atmosphere can by partially diagnosed. For instance, heatwave reinforcements by occurring droughts (Seneviratne et al., 2010; Zscheischler and Seneviratne, 2017) as well as soil moisture effects on precipitation events (Guillod et al., 2015) may be assessed by calculating imbalances in the energy budget.

## 6   Conclusions

Ecosystem experiments investigating climate change responses require a holistic, realistic climate forcing, reflecting not only the changes in the mean climate, but also representing physically consistent natural variability and changes in extreme events. To this extent, we presented a new methodology for generating climate forcing using a single Regional Climate Model (RCM) simulation, and subsequently applied it on the UHasselt Ecotron Experiment. To account for co-variances between variables and to fully capture the climate variability including extreme events, we selected an RCM simulation from the EURO-CORDEX



ensemble based on the following criteria: (i) high skill in the local present-day climate and (ii) representative of local changes in the multi-model mean.

Based on a thorough evaluation of four key variables (temperature, precipitation, relative humidity and wind speed), we found that there is no single RCM-GCM combination outperforming all others for all considered variables and metrics. We made a selection of the six best performing simulations as potential candidates and verified whether they represent the multi-model mean for the considered variables. As precipitation is considered the most important variable in ecosystem experiments, and as most GCM downscalings have large bias for this variable, we use the precipitation bias as the decisive factor to single out the simulation which will serve as forcing: CCLM4-8-17 driven by EC-EARTH.

The ecotron units are forced with climate conditions along a Global Mean Temperature (GMT) anomaly gradient, representing conditions of a 0°C (historical), +1°C (present-day), +1.5°C, +2°C, +3°C and +4°C warmer world. Five-year time windows corresponding to these warming levels are defined based on when the 30-year averaged GMT anomaly of EC-EARTH, the driving GCM, crosses these temperature thresholds. Subsequently, the ecotron forcing is extracted from the 3-hourly RCM simulation according to the time windows.

The UHasselt Ecotron experiment allows to quantify and assess the ecosystem responses on changing climatic conditions, thereby accounting for the co-variances between climatic variables and their change in variability, well representing possible compound events. By applying a gradient approach, thresholds and possible tipping points can be identified.

*Code and data availability.* Reference station data of the European Climate Assessment and Dataset is publicly available at https://www.ecad.eu/. The greenhouse gas concentrations as prescribed by RCP 8.5 are available at https://tntcat.iiasa.ac.at/RcpDb/. Data from the Coordinated Regional Climate Downscaling Experiment (CORDEX) Africa framework is available at http://cordex.org/data-access/esgf/. The scripts used in the analysis are available on github: https://github.com/VUB-HYDR/2019_Vanderkelen_etal_BG.





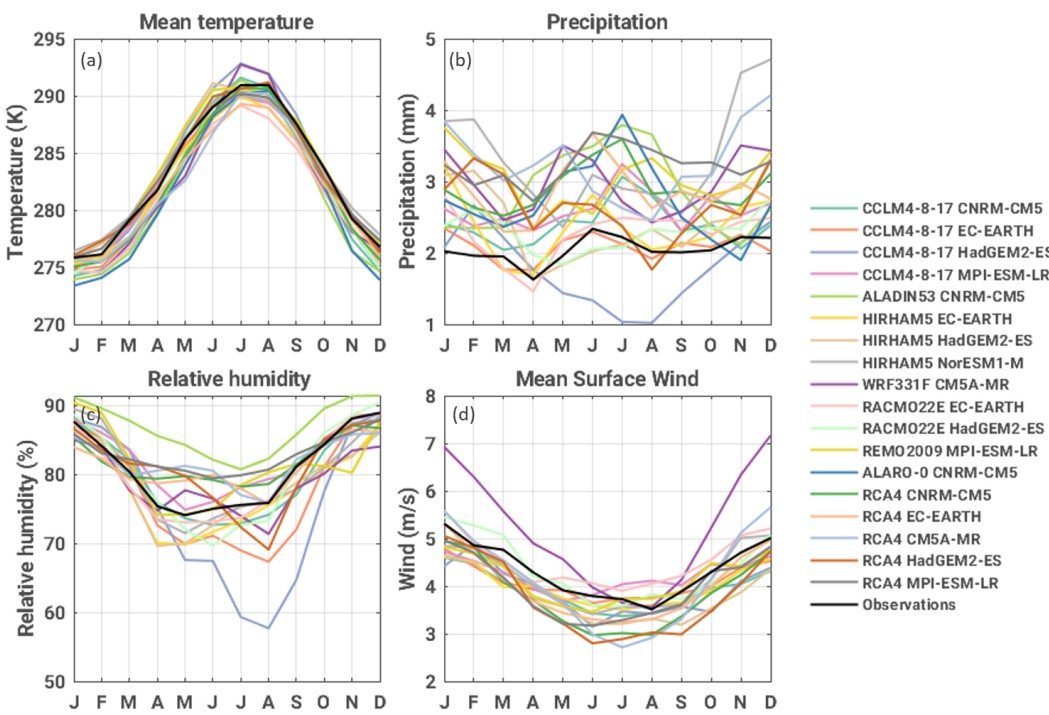

**Figure A1. Seasonal cycle of the GCM downscalings** for mean temperature (a), precipitation (b), relative humidity (c) and mean surface wind speed (d).



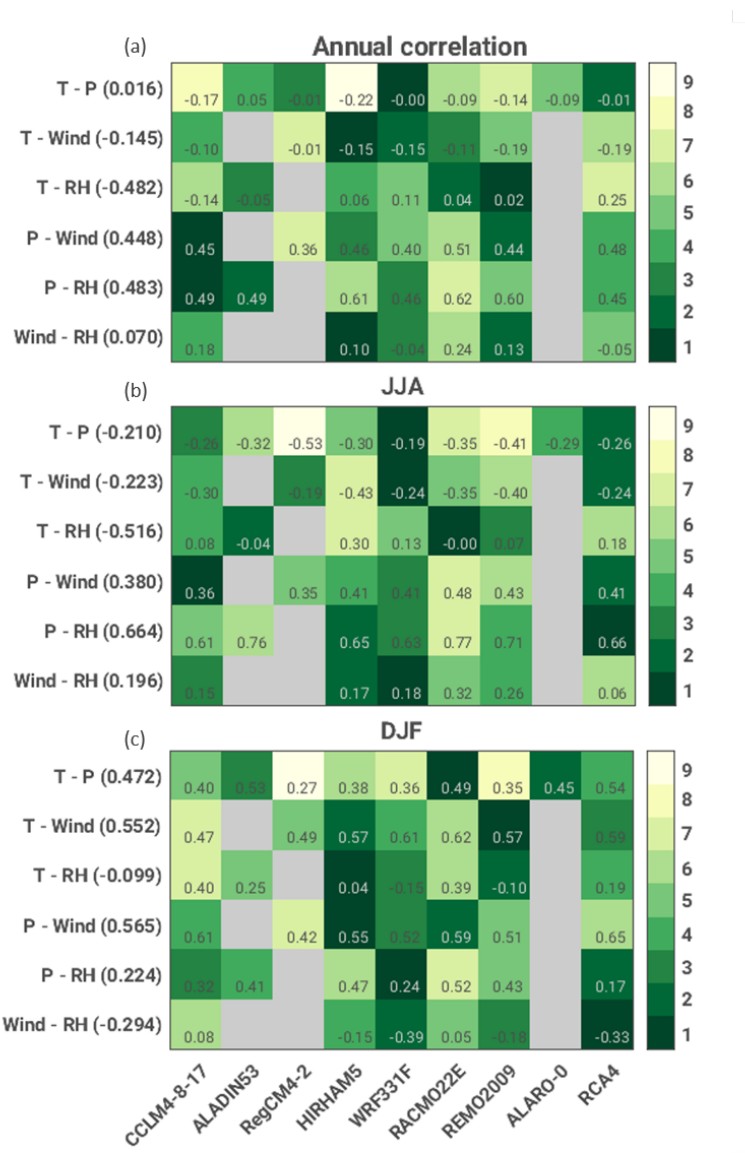

**Figure A2.** Correlations for the reanalysis downscalings (1990-2008): Annual correlations (a), correlations in June, July and August (JJA; b) and correlations in December, January and February (DJF; c). T stands for temperature, P for precipitation and RH for relative humidity. The values in the y-axis labels are the observed correlations, and the other values correlations between the simulated variables. Rankings are from 1-best to 9-worst.





**Figure A3.** Correlations for the GCM downscalings (1951-2005): Annual correlations (a), correlations in June, July and August (JJA; b) and correlations in December, January and February (DJF; c). T stands for temperature, P for precipitation and RH for relative humidity. The values in the y-axis labels are the observed correlations, and the other values correlations between the simulated variables. Rankings are from 1-best to 9-worst.



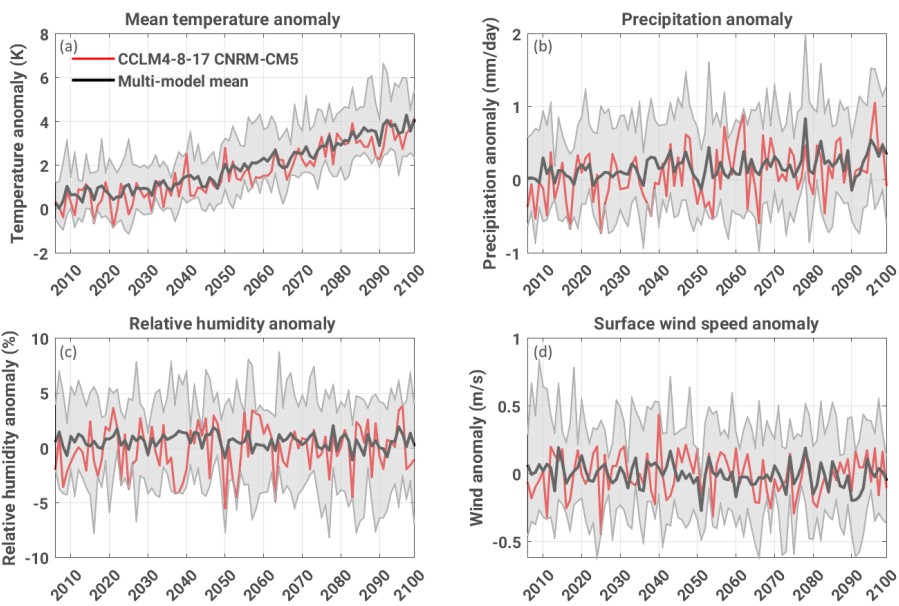

**Figure A4.** Same as Fig. 8, but now for CCLM4-8-17 CNRM-CM5.

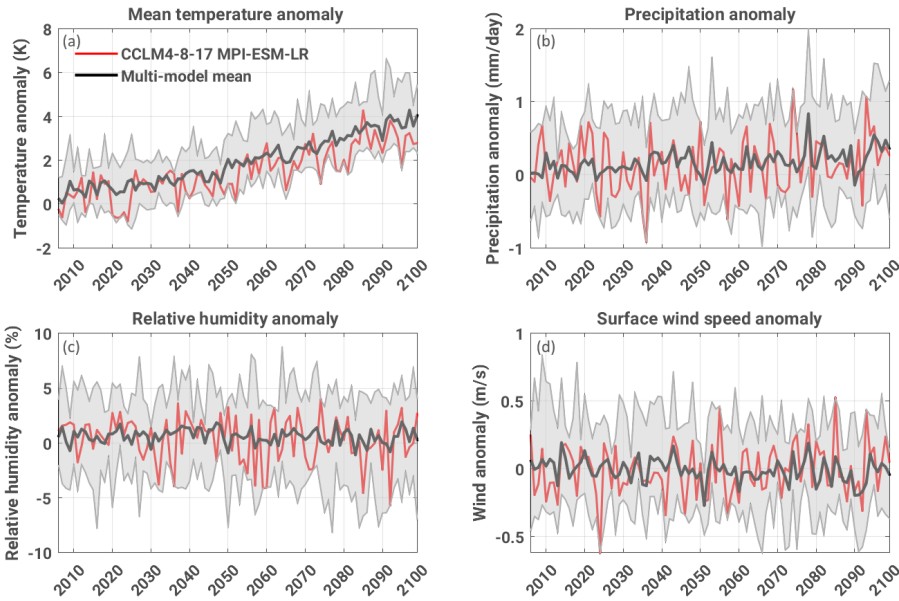

**Figure A5.** Same as Fig. 8, but now for CCLM4-8-17 MPI-ESM-LR.

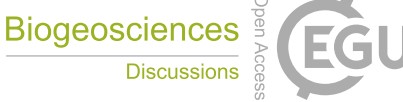

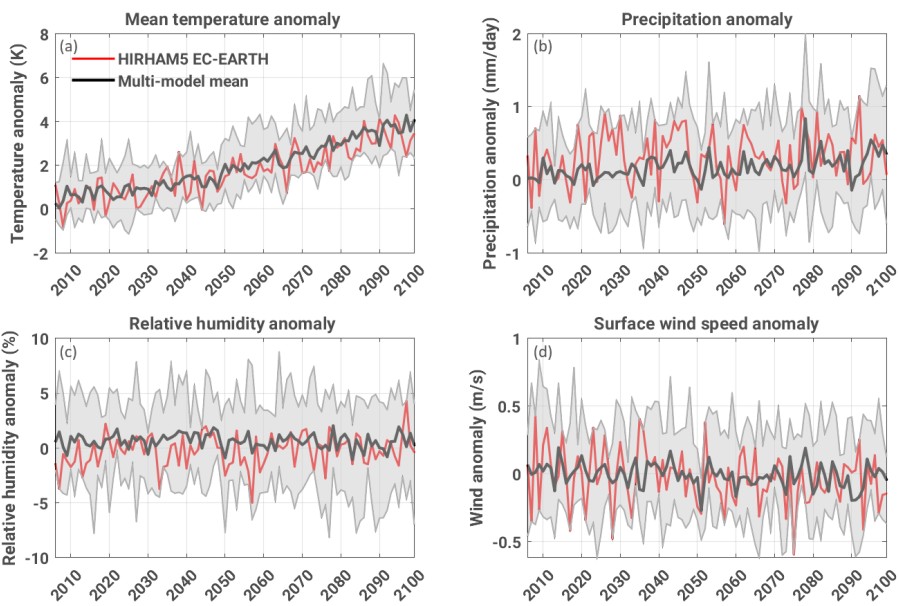

**Figure A6.** Same as Fig. 8, but now for HIRHAM5 EC-EARTH.

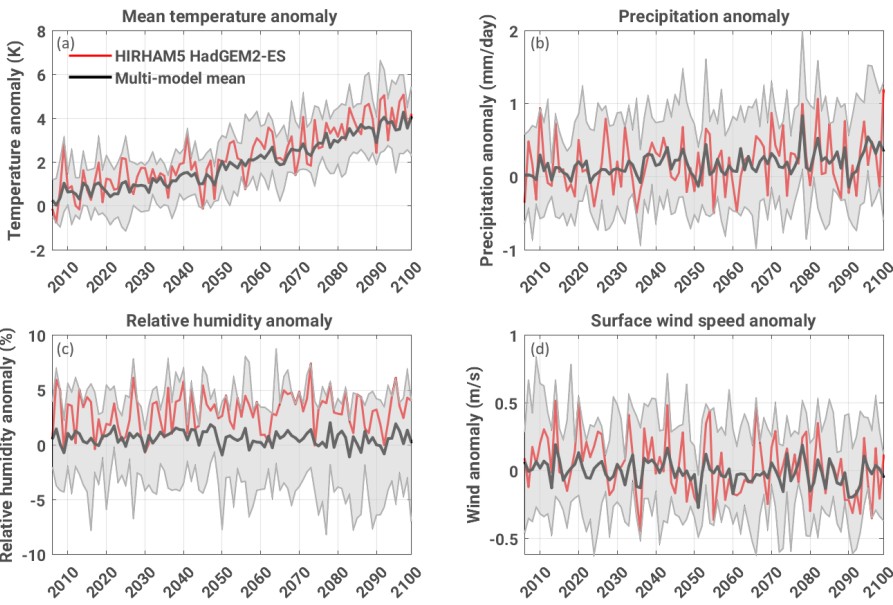

**Figure A7.** Same as Fig. 8, but now for HIRHAM5 HadGEM2-ES.





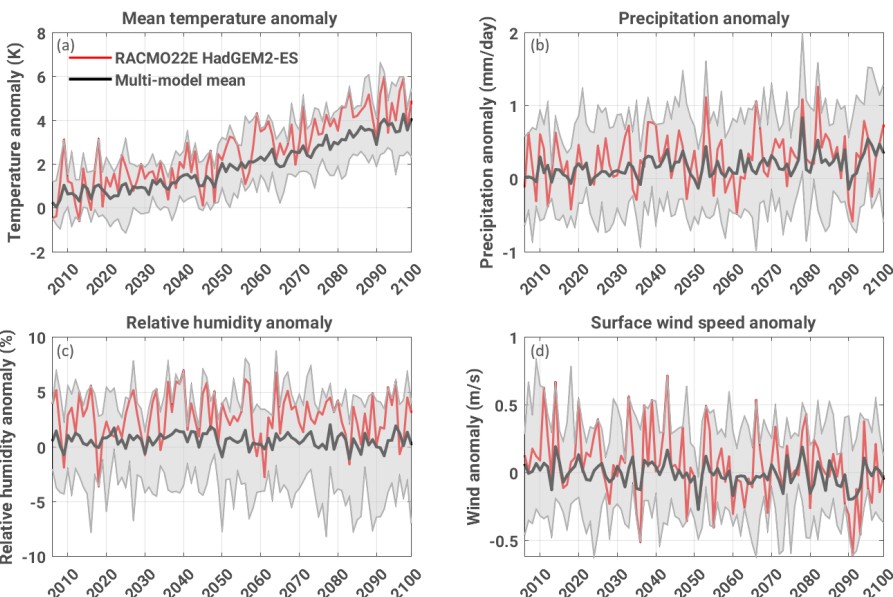

**Figure A8.** Same as Fig. 8, but now for RACMO22E HadGEM2-ES.



*Author contributions.* JK, LG, FR, WT, NB and JK conceived the ideas and designed the methodology, KK provided the 3-hourly simulation data. IV and WT led the writing of the manuscript, with major contributions from JK and FR and input from all other authors. All authors critically revised the draft and gave final approval for publication.

*Competing interests.* The authors declare that they have no conflict of interest.

5   *Acknowledgements.* Inne Vanderkelen is a research fellow at the Research Foundation Flanders (FWOTM920). Wim Thiery was supported by an ETH Zurich postdoctoral fellowship (Fel-45 15-1). The Uniscientia Foundation and the ETH Zurich Foundation are thanked for their support to this research. We are grateful to the World Climate Research Programme (WRCP) for initiating and coordinating the EURO-CORDEX initiative, to the modelling centres for making their downscaling results publicly available through ESGF. Computational resources and services were provided by the Shared ICT Services Centre funded by the Vrije Universiteit Brussel, the Flemish Supercomputer Center

10   (VSC) and FWO. The authors thank the Flemish government (through Hercules stichting big infrastructure and the Fund for Scientific Research Flanders project G0H4117N), LSM (Limburg Sterk Merk, project 271) for providing funds to build the UHasselt Ecotron; Hasselt University for both funding and policy support (project BOF12BR01 and Methusalem project 08M03VGRJ); and the ecotron research committee for useful comments on the experimental design. We also thank RLKM (Regional Landscape Kempen and Maasland) for its collaboration and support."





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
