# Peer review of "A new approach for assessing climate change impacts in ecotron experiments"

_Biogeosciences, 2019_

## Referee Comment (RC1) · Anonymous Referee #1 · 1 Oct 2019

This review focuses mainly on the design and evaluation of the UHasselt Ecotron Experiment, as was requested by the editor. However, I did read through the entire manuscript.

I am unclear about the three compartments that the authors refer to when describing their ecotron facility. They cite Rineau et al (in review) which apparently describes these, and other essential details (e.g., like which ecosystem processes will be measured and how they will be measured). I have no access to this paper.

Regardless, there is sufficient information in the methods of the manuscript to give me pause and concern. With 12 ecotron units, and what looks to be 12 individual treatment combinations, it appears that only one macrocosm will be used per treatment combination, with no experimental replication. This looks like a so-called "regression design".

These designs are fine. However, the absence of spatial replication makes it essential to obtain robust baseline ecosystem response conditions under "control" conditions (i.e. the conditions under which the control macrocosm in the spatially-unreplicated experiment will be maintained). A robust baseline for a multi-year study would require using the first year of the study to obtain/quantify the particular "behavioral personalities" of each individual (each of the 12) macrocosm. Then, only once each "personality" has been measured, can a rigorous assessment of treatments be reliably measured in the following 4-5 years. Without such a pre-assessment, it will be impossible to know whether treatment responses–evaluated against a single "control" macrocosm–are due to the treatment(s) or to an anomalous "control macrocosm" (analogous to a random "crazy personality"). This is a really critical need, and critical shortfall in the study design, as I understand it, and should be addressed. Perhaps I missed this, but I also did not see any description of how the empirical data collected from the 12-ecotron experiment would be statistically analyzed, nor did I see any specific research questions or hypotheses articulated.

I'm wondering whether the problem of the lack of spatial replication could be addressed by reducing the number of treatment combinations to six, so that there would be at least two replicate ecotron units per treatment combination.

I do appreciate the approach of using data from downscaled climate models to guide which experimental treatments to include. I also like the use of real-time ICOS data to incorporate realistic climate variability to some of the treatments. It is my understanding that these models deliver daily (24 h means or sums) resolution data, that would not be suitable to understand sub-daily/diel climate/weather variability. Is that what the ICOS data will be used for? It would certainly be important to retain diel air T, RH, and precipitation patterns in the experiment.

Taken together, the paper on its own left me with many unanswered questions. These may be covered in the Rineau et al. manuscript. I would recommend placing the essentials of that paper in the next version of this paper, particularly items that address

the questions and the issues I have identified above. Thus, based mainly on the section of the manuscript on which I was asked to focus, I feel compelled to rate the decision as "reject" at this stage of the manuscript. I would encourage improving the ms. and resubmitting, with the managing editor's approval.

──────────────────────────────

---

## Referee Comment (RC2) · Michael Bahn (Referee) · 28 Oct 2019

Dear Dr. Vanderkelen,

it has been extremely difficult to find reviewers for your manuscript, so to make some progress at this point I have decided to provide a review myself.

For several decades controlled environment facilities have been a key approach for studying effects of climate change on plants and small stature ecosystems, and since the 1990ies ecotrons and their application have been repeatedly described (e.g. Lawton et al. 1993 and 1996, Griffin et al. 1996). There is no doubt that phytotrons and ecotrons are state-of-the-art tools, whose technical capacities, including the controlled volume and the precision, have increased tremendously during the past 15 years. Such

infrastructures provide an outstanding possibility to test for individual and interactive effects of multiple global change drivers, and / or to simulate specific scenarios projected by climate models, and there is no doubt that studies based an ecotrons will yield major novel scientific insights. However, from my perspective there is only limited novelty in the description of the facility itself. For this reason, such descriptions have previously been included in the supplements of papers reporting on the actual outcome of the climate manipulation experiments performed (e.g. Arnone et al. 2008, Roy et al. 2016).

From my own background I cannot judge the degree of novelty contained in your new methodology for generating climate forcing using a single Regional Climate Model Simulation, which is one of the reasons why I sent your manuscript out for review. One of the experts on the topic, who I trust, declined my invitation to review your manuscript with the comment "This paper does not look very interesting to me - it merely describes the plan to regulate controlled environments following some very specific climate change predictions."

Taken together, I am therefore not convinced that your manuscript is advancing the field to a sufficient extent to be acceptable for publication as a full paper in Biogeosciences. I may nevertheless revise my opinion in case you manage to convince me otherwise in your author responses.

Best regards,

Michael Bahn (Editor)

---

## Author Comment (AC1) · 30 Jan 2020

**A new approach for assessing climate change impacts in ecotron experiments**

**Biogeosciences**

January 30, 2020

Inne Vanderkelen1, Jakob Zschleischler2, 3, Lukas Gudmundsson4, Klaus Keuler5, Francois Rineau6, Natalie Beenaerts6, Jaco Vangronsveld6, 7, Sara Vicca8, Wim Thiery1, 4 inne.vanderkelen@vub.be

1 Department of Hydrology and Hydraulic Engineering, Vrije Universiteit Brussel, Pleinlaan 2, 1050 Brussels, Belgium.

2 Climate and Environmental Physics, University of Bern, Bern, Switzerland.
 3 Oeschger Center for Climate Change Research, University of Bern, Bern, Switzerland.

4 Institute for Atmospheric and Climate Science, ETH Zurich, Zurich, Switzerland.

5 Department of Environmental Meteorology, Brandenburg University of Technology Cottbus-Senftenberg, Cottbus, Germany.

6 Centre for Environmental Sciences, UHasselt, Hasselt, Belgium.

7 Department of Plant Physiology, Faculty of Biology and Biotechnology, Maria Curie-Sklodowska University, Lublin, Poland.

8 Department of Biology, University of Antwerp, Wilrijk, Belgium.

**Contents**

| 1 | Reviewer 1           | 2  |
|---|----------------------|----|
|   | Reviewer 1 Comment 1 | 2  |
|   | Reviewer 1 Comment 2 | 2  |
|   | Reviewer 1 Comment 3 | 6  |
|   | Reviewer 1 Comment 4 | 9  |
|   | Reviewer 1 Comment 5 | 11 |
|   | Reviewer 1 Comment 6 | 11 |
|   | Reviewer 1 Comment 7 | 12 |
| 2 | Reviewer 2           | 13 |
|   | Reviewer 2 Comment 1 | 13 |

**Abstract**

This response letter contains numbered figures and references to these figures. To prevent confusion, the figures embedded within this response letter are called illustrations. Finally, the following convention is applied to denote modification in the original manuscript: new text.

**1 Reviewer 1**

**Reviewer 1 Comment 1**

This review focuses mainly on the design and evaluation of the UHasselt Ecotron Experiment, as was requested by the editor. However, I did read through the entire manuscript.

**Response**

We thank Reviewer 1 for his/her review of the study and the UHasselt Ecotron Experiment and the important issues raised. We understand the confusion on the main objective of this paper, which is not describing the UHasselt Ecotron experiment, but to depict a new methodology to provide climate forcing to ecosystem experiments. To this extent, the UHasselt ecotron experiment serves as an example application of the new methodology. We now clarified the objectives in the manuscript:

In this paper, we present a protocol for creating realistic climate forcing for manipulation experiments.

[...]

In our methodology, variability and co-variance between variables is preserved by selecting the best performing RCM simulation and subsequently extract the required variables from the grid cell covering the location of the experiment. By extracting a single grid cell of a single RCM simulation, climate extremes are not smoothed and the climate variability inherent to the model is fully preserved.

Below, we carefully address every comment and explain the corresponding changes in the manuscript.

Reviewer 1 Comment 2

I am unclear about the three compartments that the authors refer to when describing their ecotron facility. They cite Rineau et al (in review) which apparently describes these, and other essential details (e.g., like which ecosystem processes will be measured and how they will be measured). I have no access to this paper.

The Rineau et al. (2019) paper appeared after this review in Nature Climate Change. The reference to this paper is updated throughout the manuscript. We also added the Rineau et al. paper and its supplementary material as an appendix to this authors' response. While this study includes a more detailed description of the facility, we summarise the most important features in our current study.

We updated figure 2 in our manuscript to include a schematic overview on the three different compartments (dome, chamber and lysimeter, illustration 1).

The ecosystem processes which will be measured are listed in figure S4 of the supplementary material of Rineau et al. (2019), and we copy the table here below for reference (illustration 6). We now added some examples of which ecosystem processes will be measured in the manuscript:

The aim of the UHasselt Ecotron experiment is to study the ecological and societal impacts of climate change, by manipulating climatic variables alone or in combination and, across a wide range of predicted values, while monitoring as many soil biota and processes as possible and to translate them into socioeconomic values using heathland as a case study (Rineau et al., 2019). Examples of measured ecosystem processes are evapotranspiration, net ecosystem exchange, CH4 and N2O emissions.

Illustration 1: The UHasselt Ecotron experiment (a; picture: Liesbeth Driessen), scheme of a unit with the three compartments and the lysimeter compartment in detail (b), and overview map with location of the infrastructure and reference weather observation stations (c).

| Compartment | Measurement                         | Measurement |
|-------------|-------------------------------------|-------------|
|             |                                     | frequency   |
|             | Air temperature *                   | 1'          |
|             | Relative humidity *                 | 1'          |
|             | Wind speed *                        | 30'         |
|             | Precipitation *                     | 10'         |
| Atmospheric | CO 2 concentration *     | 30'         |
| Atmospheric | CH 4 concentration       | 30'         |
|             | N 2 O concentration      | 30'         |
|             | Air pressure                        | 30'         |
|             | Incoming / reflected radiation      | 1'          |
|             | Photosynthetically active radiation | 1'          |
|             | Temperature *                       | 10'         |
|             | Water tension *                     | 10'         |
|             | Water content                       | 10'         |
| Soil        | Electrical conductivity             | 10'         |
| 3011        | Pore CO2                            | 10'         |
|             | Weight                              | 1'          |
|             | Water leaching                      | 1'          |
|             | (suction cups for water sampling)   | 2 weeks     |

Illustration 2: Controlled and measured parameters in one macrocosm. The asterix marked variables are both controlled and measured, the others only monitored. Directly taken from Rineau et al. (2019), supplementary materials.

**Reviewer 1 Comment 3**

Regardless, there is sufficient information in the methods of the manuscript to give me pause and concern. With 12 ecotron units, and what looks to be 12 individual treatment combinations, it appears that only one macrocosm will be used per treatment combination, with no experimental replication. This looks like a so-called "regression design". These designs are fine. However, the absence of spatial replication makes it essential to obtain robust baseline ecosystem response conditions under "control" conditions (i.e. the conditions under which the control macrocosm in the spatiallyunreplicated experiment will be maintained). A robust baseline for a multi-year study would require using the first year of the study to obtain/quantify the particular "behavioral personalities" of each individual (each of the 12) macrocosm. Then, only once each "personality" has been measured, can a rigorous assessment of treatments be reliably measured in the following 4-5 years. Without such a pre-assessment, it will be impossible to know whether treatment responsesâevaluated against a single "control" macrocosmâare due to the treatment(s) or to an anomalous "control macrocosm" (analogous to a random "crazy personality"). This is a really critical need, and critical shortfall in the study design, as I understand it, and should be addressed.

**Response**

The experiment set-up follows indeed a "regression design". Like the reviewer commented, small initial differences in small-scale soil heterogeneity between different units can increase to the point that the unit becomes statistically different from the others (Rineau et al., 2019). As the reviewer correctly points out, this can prevent interpreting results from individual units like robust baseline ecosystem responses, as there is no replication of the experiment. Therefore, a cluster analysis has been conducted, which quantifies the natural variation of the 14 measured variables between the Ecotron units during 11 months prior to the experiment (illustration 3, from Rineau et al. (2019) supplementary material). Based on this analysis, the units are distributed in two gradient classes, minimizing the natural variance (noise). The resulting unit distribution over the two gradients is illustrated in Rineau et al. (2019) by figure 1 (see illustration 4 below).

We thank the reviewer for pointing out that our manuscript is lacking this information. To accommodate this, we updated the manuscript as follows:

The climatology of the unit forced by +1°can thereby be directly compared to the unit driven by the ICOS station and thus representing the present-day observed conditions. In this regression design, there is no experiment replication. To minimize the noise in initial ecosystem responses, the units are allocated to the two gradient experiments based on a cluster analysis of the variance of the 14 variables measured during a test period of 11 months (Rineau et al., 2019).